# Automatic P-Phase-Onset-Time-Picking Method of Microseismic Monitoring Signal of Underground Mine Based on Noise Reduction and Multiple Detection Indexes

**DOI:** 10.3390/e25101451

**Published:** 2023-10-16

**Authors:** Rui Dai, Yibo Wang, Da Zhang, Hu Ji

**Affiliations:** 1BGRIMM Technology Group Co., Ltd., Beijing 102628, China; 2Institute of Geology and Geophysics, Chinese Academy of Sciences, Beijing 518055, China; 3University of Chinese Academy of Sciences, Beijing 100049, China; 4China-South Africa Joint Research Center for Exploitation and Utilization of Mineral Resources, Beijing 102628, China; 5China-South Africa BRI Joint Laboratory for Sustainable Development and Utilization of Mineral Resources, Beijing 102628, China

**Keywords:** microseisms, P-phase onset time picking, STA/LTA method, AIC method, skew and kurtosis method, wavelet coefficient threshold denoising in time–frequency domain

## Abstract

The underground pressure disaster caused by the exploitation of deep mineral resources has become a major hidden danger restricting the safe production of mines. Microseismic monitoring technology is a universally recognized means of underground pressure monitoring and early warning. In this paper, the wavelet coefficient threshold denoising method in the time–frequency domain, STA/LTA method, AIC method, and skew and kurtosis method are studied, and the automatic P-phase-onset-time-picking model based on noise reduction and multiple detection indexes is established. Through the effect analysis of microseismic signals collected by microseismic monitoring system of coral Tungsten Mine in Guangxi, automatic P-phase onset time picking is realized, the reliability of the P-phase-onset-time-picking method proposed in this paper based on noise reduction and multiple detection indexes is verified. The picking accuracy can still be guaranteed under the severe signal interference of background noise, power frequency interference and manual activity in the underground mine, which is of great significance to the data processing and analysis of microseismic monitoring.

## 1. Introduction

The development of mineral resources in China has gradually entered the stage of deep mining. Rock burst, collapse and other ground pressure disasters caused by deep high stress have become a major hidden danger restricting mine safety production. Microseismic monitoring technology is an important means of ground pressure risk prediction [1,2,3,4,5,6,7,8,9,10,11]. For the microseismic monitoring system, the automatic identification and accurate picking of the P-phase onset time of microseismic signals of the underground mine directly affects the timeliness and accuracy of microseismic source location, source mechanism inversion, and rock stability analysis [11,12,13]. Therefore, the automatic identification and accurate picking of the P-phase onset time of microseismic signals underground mine is the core premise of microseismic monitoring technology. Only by accurately identifying P-phase onset time can the accuracy of microseismic source location and source mechanism inversion information be guaranteed, and then the ground pressure risk underground mine can be assessed [7,14,15] . 

The background noise, power frequency interference, artificial activity and other signals of the underground mine will seriously interfere with the effective microseismic signal, greatly reducing the signal-to-noise ratio of the microseismic information, resulting in increased difficulty in P-phase onset time picking. The microseismic monitoring system collects a large number of microseismic events every day. Although the P-phase onset time can be intuitively picked up by the manual operation of the data processor, the quality is greatly affected by the experience and subjectivity of the data processor [16,17].

How can we ensure the accuracy of the automatic P-phase onset time picking of low-SNR microseismic signal of the underground mine under severe interference environment? The traditional automatic P-phase onset time picking is mainly divided into the following categories: The first is the short-long-window ratio method (STA/LTA) [18,19], using the ratio of waveform amplitude, energy, waveform envelope function and other characteristic functions in the long time window and the short time window as the judgment criterion for P-phase onset time picking. The second is to assume that noise signals and microseismic signals have different statistical properties, and to use characteristic functions such as higher-order statistics (kurtosis and skewness) and entropy to realize automatic identification of microseismic signals [20,21,22]. The third type is to use cross-correlation theory to match the target waveform with the template waveform to obtain the relative P-phase onset time [23], and then add the absolute arrival time with the template waveform to obtain the P-phase onset time of the target waveform. Although the above method has effectively realized the automatic P-phase onset time picking, the picking accuracy still cannot meet the needs of microseismic location. Therefore, some scholars improve the recognition and picking effect of microseismic signals by analyzing the various characteristic differences between microseismic signals and environmental noise. For example, Song, W.Q. and Lv, S.C. combined wavelet decomposition with Akaike information criteria to improve the P-phase onset time picking accuracy of microseismic signals [24]. Tan, Y.Y.; Yu, J. and Feng, G. and Chuan, H.E made comprehensive use of the differences between microseismic signals and noise in energy, polarization and statistics, and effectively improved the anti-noise ability and picking accuracy of the P-phase-onset-time-picking method to a certain extent [25]. Some researchers also use the Time–frequency Transform (TFT) to transform the seismic wave signal in the one-dimensional time domain into the two-dimensional time–frequency domain, so that the location of the microseismic signal can be clearly identified on the time–frequency spectrum, and then signal processing and denoising can be carried out. Wavelet coefficient threshold denoising method in time–frequency domain has been widely concerned due to its simple principle [26,27] and remarkable denoising effect [10,28,29,30,31,32].

In this paper, the real-time processing of microseismic data is combined with the time–frequency domain wavelet coefficient threshold denoising method, STA/LTA method and AIC method [19,22], and the skew and kurtosis method based on higher-order statistics, and this combination establishes a P-phase-onset-time automatic picking model based on noise reduction and multiple detection indexes. Through the effect analysis of microseismic signals collected by microseismic monitoring system of coral Tungsten Mine in Guangxi, automatic P-phase onset time picking is realized. the reliability of the P-phase-onset-time-picking method proposed in this paper based on noise reduction and multiple detection indexes is verified. The picking accuracy can still be guaranteed under the severe signal interference of background noise, power frequency interference and manual activity of the underground mine, which is of great significance to the data processing and analysis of microseismic monitoring.

## 2. Data Collection

Taking the coral tungsten mine of Guangxi Guihuacheng Co., Ltd. (Hezhou, China), as the research object, the coral tungsten mine of Guangxi is located in the territory of Coral Town, Zhongshan County, and its geographical center coordinates are 111°21′13″ east longitude, 24°20′02″ north latitude. A total of 40 microseismic monitoring stations are arranged in the goaf or around the stope in the middle section of 75 m, 35 m and −5 m, among which 16 monitoring points are arranged in the middle section of 75 m, respectively, at 0# perforation, 2# perforation, 4# perforation, 3# perforation, 5# perforation and 7# perforation. A total of eight monitoring points were arranged in the middle section of 35 m, which were arranged at 0# perforation, 1# perforation and 3# perforation. A total of 16 monitoring points were arranged in the middle section of −5 m, which were arranged in 0# perforation, 2# perforation, 4# perforation, 3# perforation, 5# perforation and 7# perforation. The layout plan of the microseismic monitoring network is shown in Figure 1, where the triangle is the microseismic station and the number is the sensor number.

Typical microseismic signals collected by the microseismic monitoring system are shown in Figure 2, and typical blasting signals are shown in Figure 3.

## 3. Method

### 3.1. Noise Reduction Method

The time–frequency analysis was carried out on the original microseismic monitoring data, and the threshold filtering of wavelet coefficient in the time–frequency domain was adopted [10] to reduce the noise wavelet coefficient. The time domain diagram and time–frequency diagram of the original signal and denoise microseismic signal underground mine are shown in Figure 4. The time domain diagram and time–frequency diagram of original signal and denoise blasting signal underground mine are shown in Figure 5. The SNR of the mine microseismic waveform was increased from 9.0768 of the original waveform to the highest 21.2767 (SNR is the ratio of the amplitude root mean square of the waveform in a short time window with the first arrival time as the center), and the SNR of mine blasting waveform was increased from 14.06782 of the original waveform to the highest 27.1316. And, the background noise in the same frequency band with the signal can be removed well, which is conducive to the recognition and accurate picking of P-phase onset time.

### 3.2. STA/LTA Method

The STA/LTA method is the most widely used automatic picking method at present. Its main principle is to pick up P-phase onset time according to the change trend of the long–short mean value of the microseismic characteristic function, give a sliding long time window, take a short time window in this window, and set the characteristic function CF(i). The P wave arrival time is judged by the ratio of the short window signal characteristic function mean (STA) and the long window signal mean (LTA). When the ratio is greater than the preset arrival threshold (Thr), the corresponding time is the P-phase onset time. STA mainly reflects the average value of microseismic signals, and LTA mainly reflects the average value of background noise. At the arrival time of the microseismic signal, STA changes faster than LTA, and the corresponding STA/LTA value will have a significant increase. When the ratio is greater than Thr, a microseismic event can be determined to occur, so as to achieve the purpose of automatic detection and picking up P-phase onset time of microseismic. The STA/LTA algorithm formula [18] is as follows:(1)$$STA(i)=\frac{1}{ns}\sum_{j=i-ns}^{i}CF(j)$$
(2)$$LTA(i)=\frac{1}{nl}\sum_{j=i-nl}^{i}CF(j)$$
where i is the sampling time, ns is the length of the short-time window, nl is the length of the long-time window, and the characteristic function is CF(i), which generally represents the amplitude and energy of the microseismic shape.

The expression of Thr(i) is
(3)$$Thr(i)=\frac{STA(i)}{LTA(i)}$$

When Thr(i) is greater than a certain threshold λ, the corresponding time is the time.

Because the data processor needs to deal with a large number of low-SNR original microseismic signals, the selection of low-SNR microseismic signals when verifying the STA/LTA method, the selection of feature function and time threshold is the key of the algorithm, which plays a decisive role in the accuracy of the picking results. CF(i) takes the square of the amplitude, when the threshold λ is 1.5, 4, and 8, respectively, and the picking effect is shown in Figure 6. The original blasting waveform is shown in Figure 6a, and the STA/LTA function when the threshold λ is 1.5, 4, and 8, respectively, is shown in Figure 6b–d. The black solid line is the manually picked P-phase onset time, the pink dashed line is the automatically picked P-phase onset time when the threshold is 1.5, and the red dashed line is the automatically picked P-phase onset time when the threshold is 4. When the threshold is 8, the P-phase onset time is not picked up; it can be seen that the accuracy of the selection of the threshold λ directly determines the quality of the STA/LTA method. Although the STA/LTA method can quickly and automatically pick up the P-phase onset time, the picking precision depends on the accuracy of the time threshold selection. In general, the short-time window length is set to 2–3 times the main period of the microseismic signal, and the long-time window length is 5–10 times the short-time window length [19]. After repeated testing and comparative analysis, the short-time window length in this paper is 100 sampling points, the long-time window length is set to 8 times the short-time window length, and the threshold λ is set to 4.

### 3.3. AIC Method

AIC is based on the concept of entropy, derived from information theory and the maximum likelihood principle, and is used to estimate the complexity of the model and measure the goodness of the model fit. Because the statistical characteristics of noise and microseismic signal are very different, the two signals have the worst fit at the junction of microseismic signal and noise, and the AIC value of this junction is the smallest, that is, the P-phase onset time of microseismic event. Given a microseismic datum Y of length N, assuming that the *I*-th point is the best boundary between the noise component and the microseismic signal component, the signal is divided into two segments at the *I*-th point, and the corresponding AIC value can be approximated as [9,22]
(4)$$AIC(i)=i\cdot lg(var\left(Y\left[1,i\right]\right))+(N-i-1)\cdot lg(var\left(Y\left[i+1,N\right]\right))+C$$
where var represents the variance of the data, Y[i + 1, N] is the amplitude of the signal in the window length N − (i + 1), and C is the constant.

The selection of low-SNR microseismic-signal-picking results shown in Figure 7a is the original microseismic signal, the black solid line is the arrival time of manually picking P-wave, the red dashed line is the location of the local minimum of AIC function, and the pink dashed line is the location of the global minimum of the AIC function; Figure 7b: The time window adopts the AIC function from the first 0.12 s to the last 0.12 s. In the case that the time window just includes the waveforms before and after the arrival of the microseismic signal, the AIC function changes at the first arrival time, and the local minimum is the global minimum, and the picking accuracy is high at that time. Figure 7c: The time window adopts the AIC function from the first 0.12 s to the last 0.83 s, and the position where the global minimum appears changes. The results show that the local minimum obtained by the AIC method is suitable for the situation where the approximate time window of the P-phase onset time is known, and can be used in combination with the STA/LTA method.

### 3.4. Skewness and Kurtosis Method

The skewness and kurtosis method is one of the important methods that measures the Gaussian characteristics of microseismic recordings. Based on the assumption that microseismic signals are non-Gaussian and background noise records are Gaussian [20], the skewness and kurtosis features of signal amplitude are used as the feature functions of the initial arrival pick-up of microseismic signals, and the P-phase onset time is picked up by using the position information of the extreme value of skewness and kurtosis, and the formula is as follows:(5)$$S(i)=\frac{1}{W}\sum_{j=i}^{i+W}{\left[\frac{Y(j)-\bar{Y}}{\sigma }\right]}^{3}$$
(6)$$K(i)=\frac{1}{W}\sum_{j=i}^{i+W}{\left[\frac{Y(j)-\bar{Y}}{\sigma }\right]}^{4}-3$$
where W time window length, $\bar{Y}$ and σ are the time window amplitude mean and standard deviation. The length of the time window W should not be too large or too small, and should try to just contain microseismic signals. The time window is too long, resulting in reduced sensitivity, and conversely, some high-amplitude noises are misidentified, reducing the picking accuracy. There is a big difference between the P-phase onset time picked by kurtosis and skewness directly and the P-phase onset time (black solid line) manually; the results are shown in Figure 8 and Figure 9.

### 3.5. Noise Reduction and Multiple Detection Indexes Method

Based on the time–frequency analysis of the original microseismic monitoring data, the wavelet coefficient threshold filtering in time–frequency domain is adopted to reduce the noise wavelet coefficient, remove the background noise in the same frequency band as the signal, improve the signal-to-noise ratio of the mine microseismic shape, and establish the P-phase-onset-time automatic picking model based on denoising and STA/LTA, AIC and PAI-S/K indicators. The time-picking effect of time–frequency domain wavelet coefficient threshold denoising combined with the STA/LTA method is shown in Figure 10. (a) is the original microseismic shape, (b) is the filtered microseismic signal, (c) is the original STA/LTA function, (d) is the filtered STA/LTA function, and the black solid line is the manually picked P-phase onset time. The pink dashed line is the automatic P-phase onset time of the original waveform, and the red dashed line is the automatic P-phase onset time pick up position of the filtered micro-shock shape. It can be seen that the accuracy of the filtered waveform pick up is greatly improved compared with the manually picked P-phase onset time (black solid line). As shown in Figure 11, (a) is the original microseismic shape, (b) is the filtered microseismic shape, (c) is the original AIC function, and (d) is the filtered AIC function. The black solid line is the manually picked P-phase onset time, the pink dashed line is the automatically picked P-phase onset time of the original waveform, and the red dashed line is the automatically picked P-phase onset time of the filtered microseismic shape. Both the original microintegral signal and the filtered microseismic signal fall into the global minimum when using the AIC function alone to pick up the P-phase onset time, and the error is larger than that when manually picking up the P-phase onset time (black solid line). As shown in Figure 12 and Figure 13, the arrival-time-picking effect of the kurtosis function and skewness functions, respectively. It can be seen that, when using the kurtosis function or skewness function alone, compared with manually picking P-phase onset time (black solid line), the P-phase onset time (red dashed line) of the filtered data is smaller than that of the original microseismic P-phase onset time (pink dashed line). However, it is still impossible to accurately pick out the P-phase onset time.

As shown in Figure 14, the P-phase-onset-time-picking results of the detection model after filtering, STA/LTA and AIC function are combined. It can be seen that the accuracy of P-phase onset time (red dashed line) of the filtered data is greatly improved compared with that of the original microseismic P-phase onset time (pink dashed line), and the problem of falling into the global minimum when AIC function is used alone is avoided. As shown in Figure 15, the P-phase-onset-time-picking effect of the detection model appears after combining filtering, STA/LTA and kurtosis K function. It can be seen that the accuracy of the P-phase onset time (red dashed line) of the filtered data is greatly improved compared with that of the original microseismic P-phase onset time (pink dashed line). As shown in Figure 16, the P-phase-onset-time-pick-up effect of the detection model after combining filtering, STA/LTA and skewness S function, and the P-phase onset time pick up accuracy is greatly improved. Another microseismic shape with lower SNR is studied; from the picking results in Figure 17, Figure 18 and Figure 19, it can be seen that the deviation of P-phase onset time of manual pick up increase with the decrease in SNR. However, the P-phase-onset-time-pick-up effect of the detection model decreased after combining filtering.

## 4. Results

The short–short window (STA/LTA) ratio method has a good effect on the recognition of P-phase onset time of the signal after noise reduction, but the automatic P-phase-onset-time-picking accuracy is not high enough, and it is difficult to select an accurate threshold, while the AIC picking method has a high local picking accuracy. Therefore, the P-phase-onset-time-picking detection model combined with filtering, STA/LTA and the AIC function is chosen as one of the picking methods in this paper. In addition, the detection model combining denoising, and STA/LTA with kurtosis K function or skewness S function can realize the precise pick up of microseismic signal P-phase onset time when the kurtosis and skewness of microseismic records meet the discrimination threshold at the same time. From the picking results in Figure 14, Figure 15, Figure 16, Figure 17, Figure 18 and Figure 19, it can be seen that the three P-phase-onset-time-picking detection models all have higher accuracy, meet the requirements of microseismic location accuracy, and can save labor costs and improve efficiency. In order to better demonstrate the accuracy of the automatic picking method, the automatic picking results of multiple microseismic signals collected by the microseismic monitoring system of the coral Tungsten Mine in Guangxi were compared with the manual picking results, and the statistical distribution of errors (automatic picking results minus manual picking results) between the results of three P-phase-onset-time detection models and the manual picking results were obtained, as shown in Figure 20: (a) for the detection model error by the combination of denoise, STA/LTA and AIC function; (b) for the detection model error by the combination of denoise, STA/LTA and kurtosis K function; (c) for the detection model error by the combination of denoising, STA/LTA and skewness S function. It can be seen from the figure that the overall picking errors of the three picking methods are small, mainly distributed in the range of −25 to 0 sampling points. Considering that the sampling rate of the data acquisition system is 6000 Hz, the corresponding time pick error is 0~4.17 ms. The first arrival result of the automatic picking method is generally smaller (time advance) than that of the manual picking, which is reasonable, because the artificial picking is generally based on the position of the amplitude maximum point, and then picks up the P-phase onset time near the position; so, the first arrival result may be too large. The deviation of P-phase onset time of manual pick up will increase with the decrease in SNR.

## 5. Conclusions

In this paper, the wavelet coefficient threshold denoising method in the time–frequency domain, STA/LTA method, AIC method, and PAI-S/K method based on skew and kurtosis are studied, and the P-phase-onset-time automatic picking model based on denoising and multiple detection indexes is established, and the P-wave arrival time’s automatic and accurate picking is realized. And, through the analysis of the P-phase-onset-time automatic picking effect of the microseismic signals collected by the microseismic monitoring system of coral Tungsten Mine in Guangxi, the conclusions and future research prospects are as follows:

(1)When the signal-to-noise ratio is low, the P-phase-onset-time pickup and detection model combined with denoising, STA/LTA and the AIC function is adopted; the P-phase-onset-time pick up and detection model combined with denoising, STA/LTA and kurtosis K function is adopted; the P-wave-time pick up and detection model combined with denoising, STA/LTA and skew S function has strong anti-interference ability. It is more reliable when picked up.(2)The STA/LTA method can quickly pick up the P-phase onset time, and the practical application is still very wide. The selection of the threshold is the key of the algorithm, and the picking accuracy is extremely dependent on the accuracy of the selection of the threshold. The AIC method relies on the correct selection of the window when calculating the local minimum point of AIC function. When the time window just contains the initial arrival of the microseismic P wave, this method has a high accuracy. This method is suitable for the situation where the approximate position of the P wave’s arrival time is known.(3)This paper proposes an automatic P-phase-onset-time-picking method based on denoising and multiple detection indexes. In the case of serious signal interference such as background noise, power frequency interference and manual activity in the mining site, the picking accuracy can still be guaranteed, which is of great significance for the processing and analysis of microseismic monitoring data.(4)There have been many machine learning models that can reliably perform the picking job at a regional/global scale, while in the seismology perspective, it is very necessary to carry out further research in the field of mine microseismic monitoring.

## Figures and Tables

**Figure 1 entropy-25-01451-f001:**
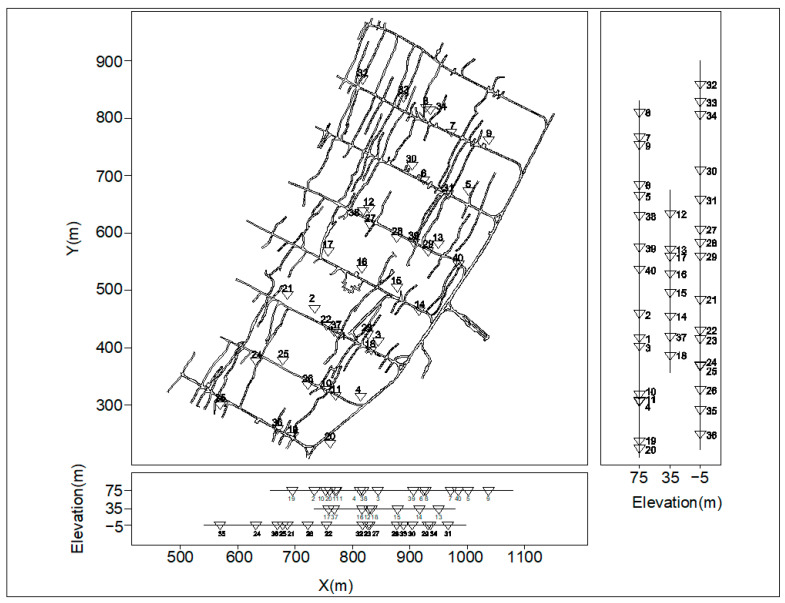
Layout plan of microseismic monitoring network for Guangxi Coral Tungsten Mine.

**Figure 2 entropy-25-01451-f002:**
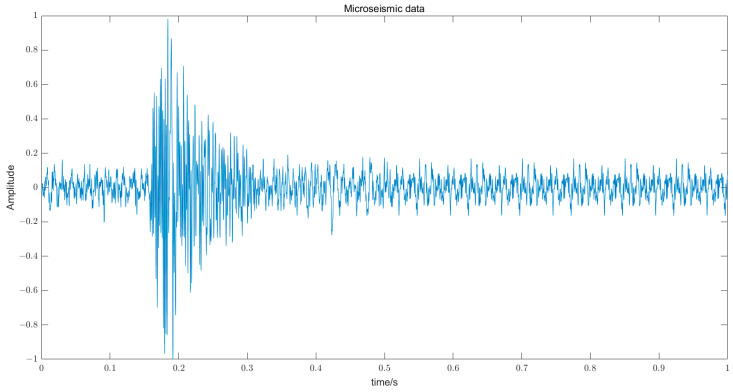
Typical microseismic signal from underground mines in Guangxi Coral Tungsten.

**Figure 3 entropy-25-01451-f003:**
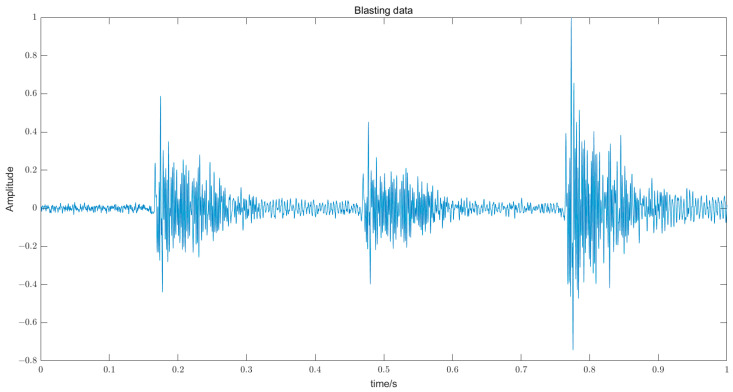
Typical blasting signal from underground mines in Guangxi Coral Tungsten.

**Figure 4 entropy-25-01451-f004:**
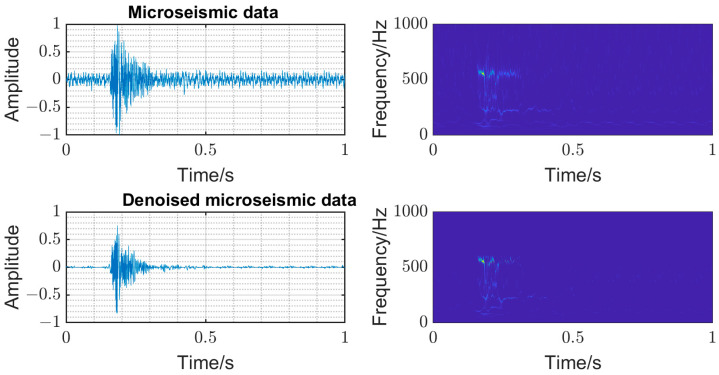
Time domain diagram and time–frequency diagram of original signal and denoise microseismic signal underground mine.

**Figure 5 entropy-25-01451-f005:**
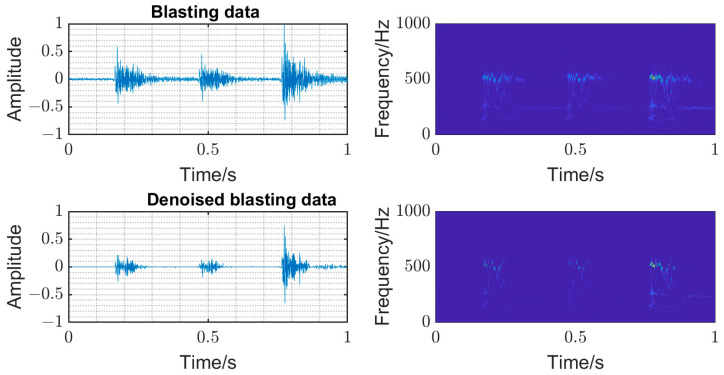
Time domain diagram and time–frequency diagram of original signal and denoise blasting signal underground mine.

**Figure 6 entropy-25-01451-f006:**
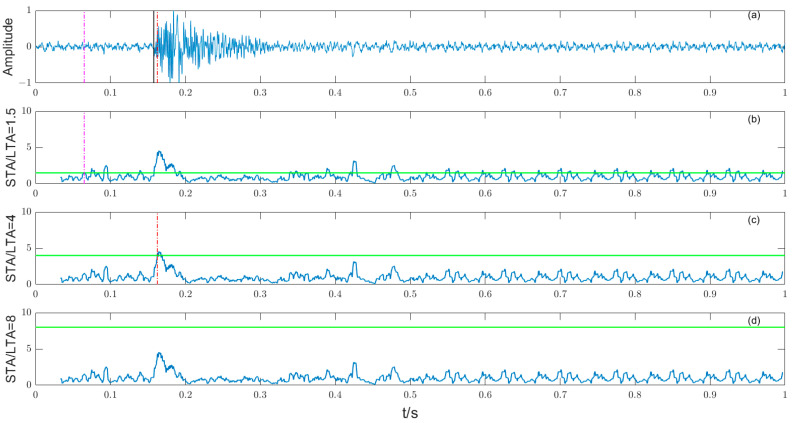
(**a**) The original microseismic signal; (**b**) STA/LTA function λ of 1.5; (**c**) STA/LTA function λ of 4; (**d**) STA/LTA function λ of 8.

**Figure 7 entropy-25-01451-f007:**
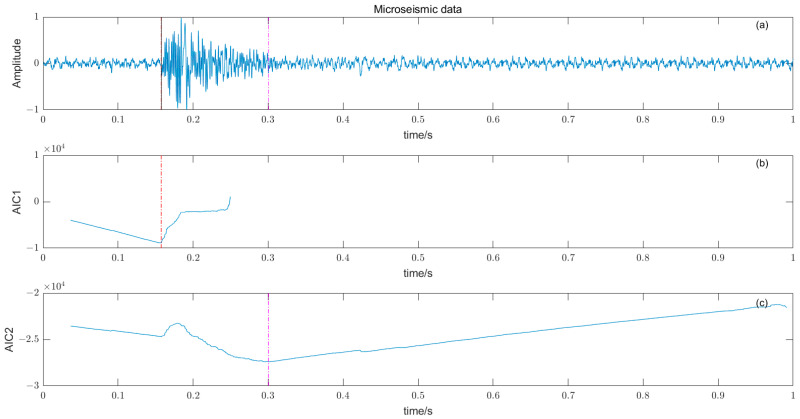
(**a**) The original microseismic signal. (**b**) The time window adopts the AIC function from first to front 0.12 s to back 0.12 s. (**c**) The time window adopts the AIC function from first to front 0.12 s to back 0.83 s.

**Figure 8 entropy-25-01451-f008:**
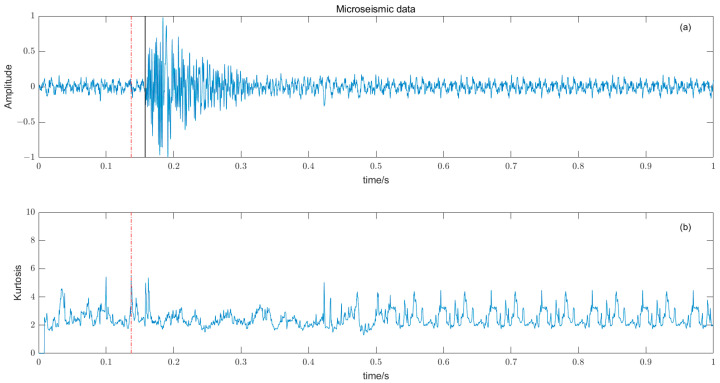
(**a**) Original microseismic signal. (**b**) Kurtosis function.

**Figure 9 entropy-25-01451-f009:**
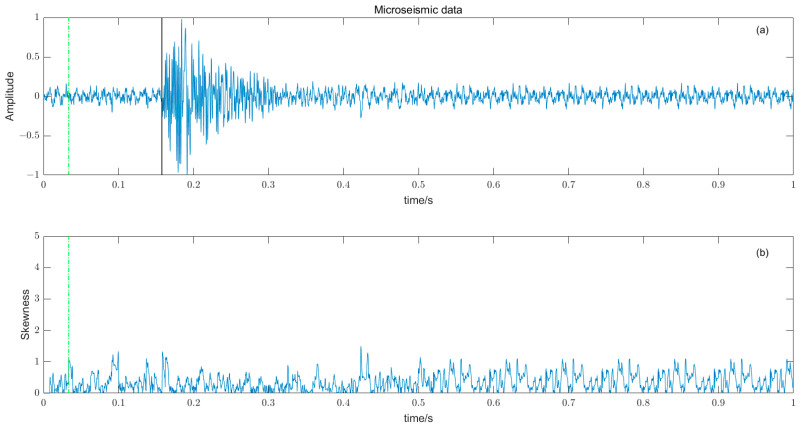
(**a**) The original microseismic signal. (**b**) Skewness function.

**Figure 10 entropy-25-01451-f010:**
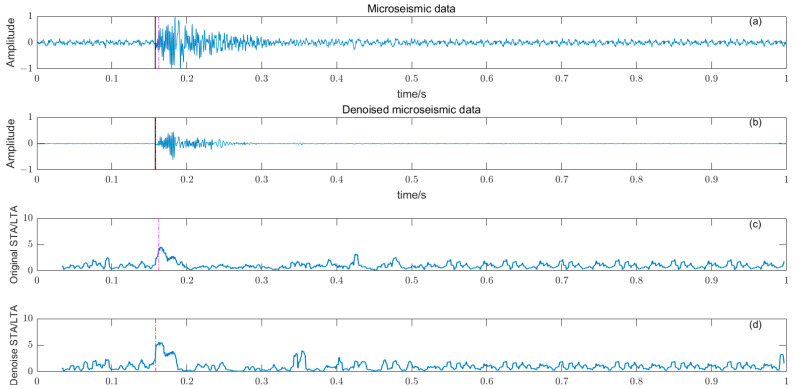
(**a**) Original microseismic signal. (**b**) Denoised microseismic signal. (**c**) Original STA/LTA function. (**d**) Denoised STA/LTA function.

**Figure 11 entropy-25-01451-f011:**
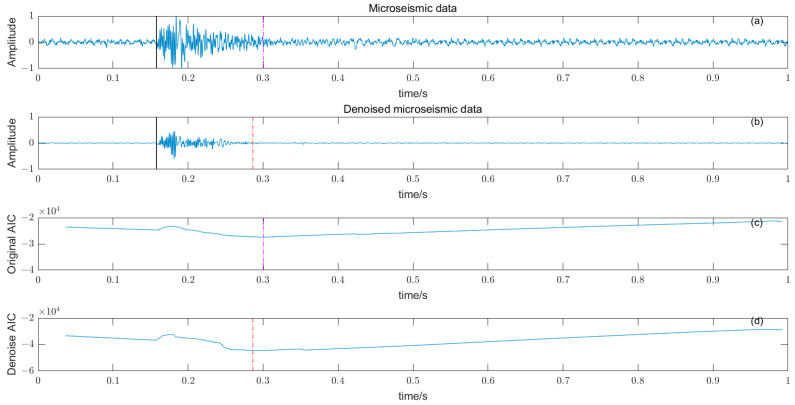
(**a**) Original microseismic signal. (**b**) Denoised microseismic signal. (**c**) Original AIC function. (**d**) Denoised AIC function.

**Figure 12 entropy-25-01451-f012:**
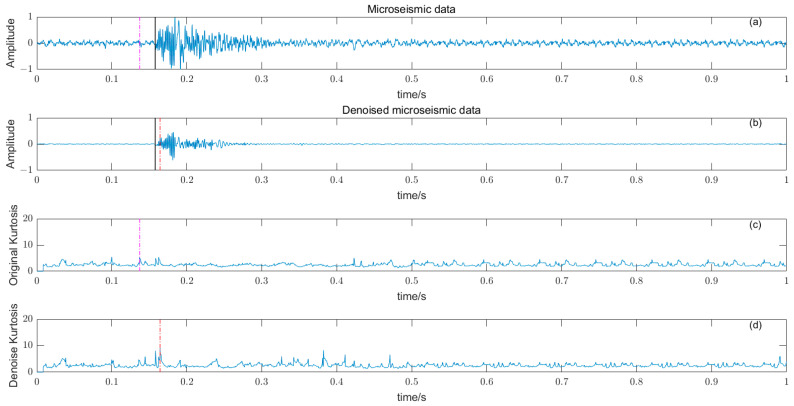
(**a**) Original microseismic signal. (**b**) Denoised microseismic signal. (**c**) Original kurtosis K function. (**d**) Denoised kurtosis K function.

**Figure 13 entropy-25-01451-f013:**
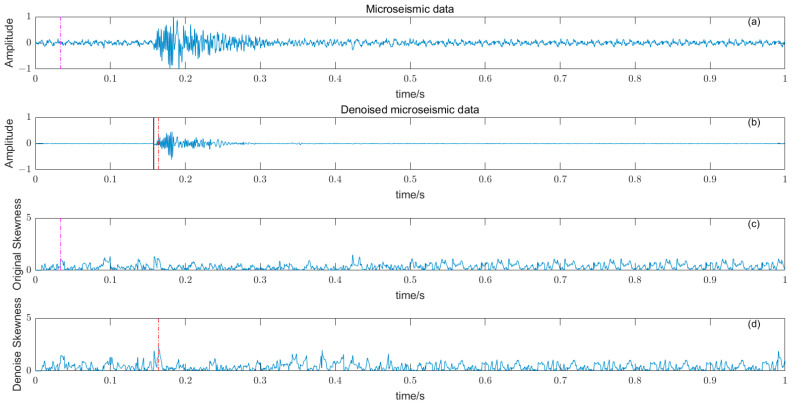
(**a**) Original microseismic signal. (**b**) Denoised microseismic signal. (**c**) Original skewness S function. (**d**) Denoised skewness S function.

**Figure 14 entropy-25-01451-f014:**
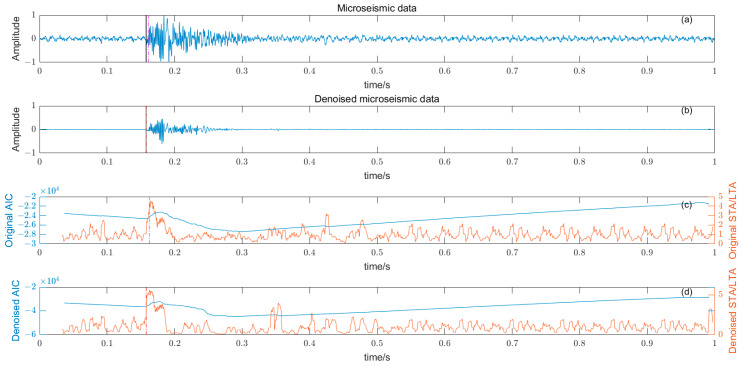
(**a**) Original microseismic signal 1. (**b**) Denoised microseismic signal 1. (**c**) Original STA/LTA-AIC function. (**d**) Denoised STA/LTA-AIC function.

**Figure 15 entropy-25-01451-f015:**
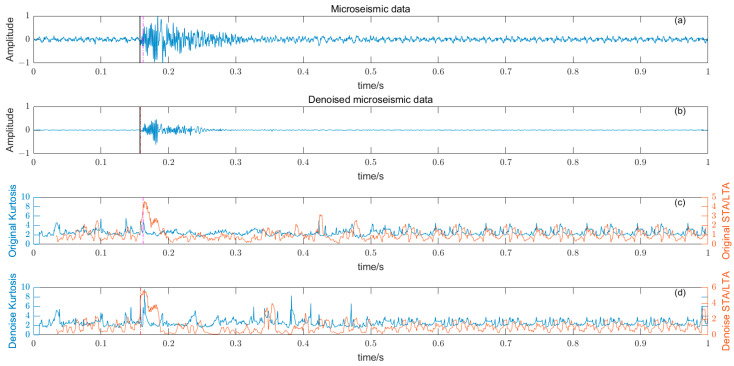
(**a**) Original microseismic signal 1. (**b**) Denoised microseismic signal 1. (**c**) Original STA/LTA-K function. (**d**) Denoised STA/LTA-K function.

**Figure 16 entropy-25-01451-f016:**
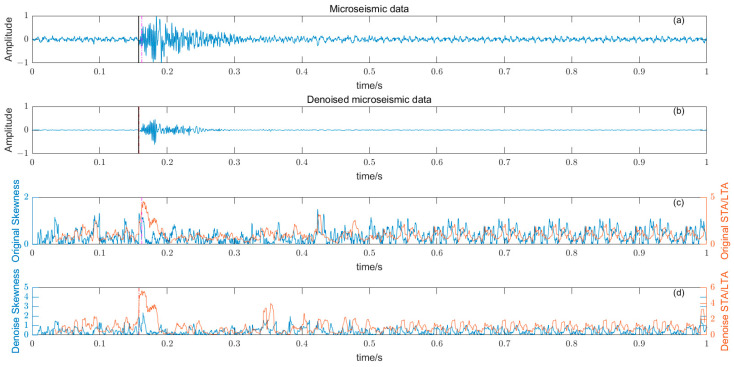
(**a**) Original microseismic signal 1. (**b**) Denoised microseismic signal 1. (**c**) Original STA/LTA-S function. (**d**) Denoised STA/LTA-S function.

**Figure 17 entropy-25-01451-f017:**
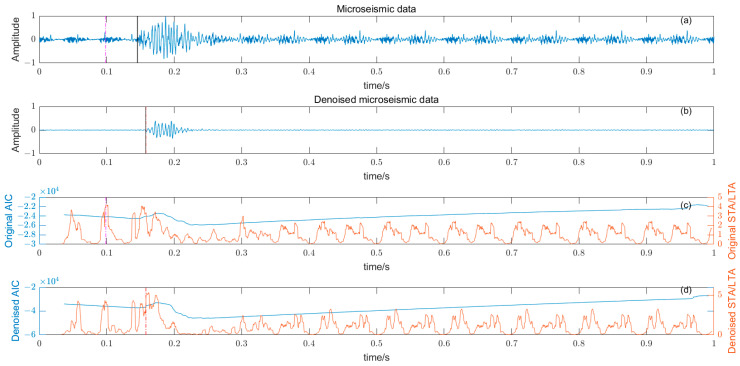
(**a**) Original microseismic signal 2. (**b**) Denoised microseismic signal 2. (**c**) Original STA/LTA-AIC function. (**d**) Denoised STA/LTA-AIC function.

**Figure 18 entropy-25-01451-f018:**
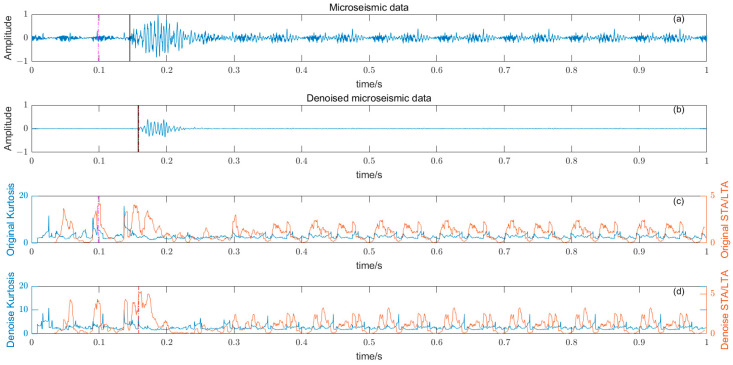
(**a**) Original microseismic signal 2. (**b**) Denoised microseismic signal 2. (**c**) Original STA/LTA-K function. (**d**) Denoised STA/LTA-K function.

**Figure 19 entropy-25-01451-f019:**
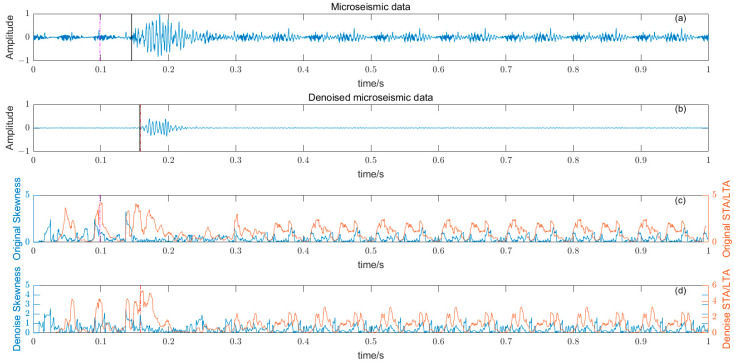
(**a**) Original microseismic signal 2. (**b**) Denoised microseismic signal 2. (**c**) Original STA/LTA-S function. (**d**) Denoised STA/LTA-S function.

**Figure 20 entropy-25-01451-f020:**
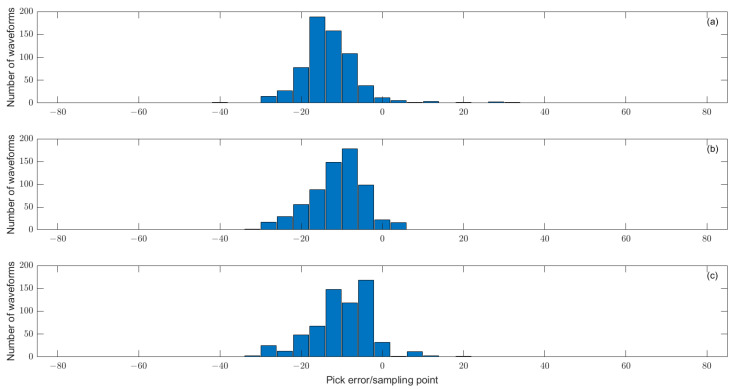
(**a**) The P-wave time picking and detecting model error combined with denoising, STA/LTA and AIC function. (**b**) The P-wave time picking and detecting model error combined with denoising, STA/LTA and kurtosis K function. (**c**) The P-wave time pick-up combined with denoising, STA/LTA and skewness S function detects the model error.

## Data Availability

Data available on request from the authors.

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
