# Peer review of "Automatic P-Phase-Onset-Time-Picking Method of Microseismic Monitoring Signal of Underground Mine Based on Noise Reduction and Multiple Detection Indexes"

_entropy, 2023, doi:10.3390/e25101451_

Round 1

Reviewer 1 Report

The work studies the impact of denoised microseismic signals on P-phase picking. I have several comments on the work for your reference:

-- The source mechanism of microseismicity in mimes could be complex (see for example, https://doi.org/10.1016/j.gete.2019.100167.  For shear-slip induced seismicity, the S energy could also be high. In such a case, it may be meaningful to mention/discuss how the S energy can affect the efficacy of the proposed method. 

-- I am sort of confused about how you actually identify P phase from high-order statistics: skewness and kurtosis. The time-varying S and K are full of extreme values. For example, Figures 8b and 9b (as well as Figures 15b and 16b) show that there are numerous local min/max values. If you don't know where is the approximate P-phase arrival, how did you pick the correct location? Is there an empirical criterion or something that you can conveniently use? Correspondingly, you should improve the presentation of P-phase picking using S and K (or more deeply, are they indeed a good/significant indicator of arrival?)

-- I would suggest using more than one example signal to validate the methodology. I think the example signal in this work is of high SNR. Including one example with lower SNR may improve the demonstration. 

-- It is not mandatory but I think there have been many machine learning models that can reliably do the picking job, even though most of them are proposed in a regional/global scale seismology perspective. Discussing/comparing with such data-driven methods can significantly improve the quality of your presentation. 

Good luck. 

A careful edit of typos may be required. 

Author Response

Thanks to the reviewer's suggestion.

Response 1: It is meaningful to discuss how S energy affects the effectiveness of the proposed method. However, the accuracy of the microseismic waveform S wave generated by mining shear slip activity is still not high even if it is manually picked up, so this article did not mention it.

Response 2: In this paper, the P-phase onset time pick up and detection model combined with denoising, STA/LTA and kurtosis K/S function is combined. STA/LTA is used to obtain the approximate time window range of the P-phase onset time, and then K/S function is used to accurately extract the the P-phase onset time.

Response 3: After modification, the pickup effect with lower SNR has been added, as shown in Figure. 17 to Figure. 19. Another microseismic shape with lower SNR is studied, it can be seen that the deviation of P-phase onset time of manual pickup increase with the decrease of SNR. But the P-phase onset time pick up effect of the detection model after combining filtering.

Response 4: There have been many machine learning models that can reliably do the picking job in a regional/global scale seismology perspective, it is very necessary to carry out further research in the field of mine microseismic monitoring. After modification, it has been added in the paper as the next research plan. 

Reviewer 2 Report

The presented topic concerns using the automatic P-phase onset time picking model method as a helpful tool in microseismic monitoring. It is a very interesting paper, but I have a few comments that should be considered before the article will be published:
1.    The Abstract and Introduction start with the same few sentences. Authors should be more creative.
2.    How many records were the presented methods tested on?
3.    There are punctuation errors (page 1, line 22, page 2, line 49, etc.).
4.    Figures 6,7 ..., and 16  should be bigger because comparing red, black and ping lines is difficult.

Author Response

Thanks to the reviewer's suggestion.

Response 1: The abstract section has been modified, please refer to lines 15 to 17 on page 1 for details.

Response 2: In this paper, 647 records were tested from Guangxin coral tungsten mine.

Response 3: Thanks to the reviewer's guidance, Punctuation errors have been corrected.

Response 4: After modification, Figures 6 to 16 in the paper have been enlarged to facilitate the identification of P-wave time of different colors.